# PAPer: Periodicity Alignment on Periodic Time Series for Forecasting

## Abstract

Time series forecasting is essential for predicting temporal dynamics across diverse domains, from meteorological patterns to urban traffic flows. Many such time series exhibit strong periodic patterns, like weekly traffic cycles, and leveraging this periodicity is crucial for forecasting accuracy. However, existing approaches typically rely on autoregressive models ($x_{t+1} = f(x_t, x_{t-1}, \dots)$) to capture these patterns implicitly or incorporate specialized modules and timestamp embeddings as auxiliary inputs explicitly. In this work, we propose PAPer: Periodicity Alignment for Periodic Time Series and demonstrate that an explicit yet simple alignment of periodic patterns without auxiliary inputs yields substantial improvements. We validate PAPer through mathematical proofs, illustrative toy examples, and extensive real-world experiments. Our results show that PAPer, when applied to state-of-the-art models, achieves performance gains of up to 7% on multiple benchmarks. Moreover, PAPer is model-agnostic and can reduce model complexity by up to 99.5% while incurring only a minor 11% performance trade-off. This work presents a foundational investigation into periodicity alignment, and the code is available at `xxx`.

## 1 Introduction

Time series forecasting-the task of predicting future values from past observations-is critical for strategic decision-making across fields like transportation, logistics, and climate science. The recent explosion of sensor data makes this capability more relevant than ever, yet it also exposes a fundamental challenge: real-world temporal data is often dynamic and chaotic.

A defining characteristic of many such time series is periodicity, where patterns recur at regular intervals. These cycles are not mere statistical artifacts but reflections of underlying system dynamics. Weekly traffic patterns inform urban planning, seasonal sales cycles drive retail inventory decisions, and daily consumption patterns are essential for power grid management. The ability to effectively capture such periodicity is crucial for taming the chaotic, dynamic nature of time series data.

Typically, forecasting models have addressed periodicity through two primary approaches: implicitly or explicitly. Implicit methods capture periodic patterns by designing models under the (linear or nonlinear) autoregressive assumption $x_t = f(x_{t-1}, x_{t-2}, \dots)$. These models include traditional approaches such as exponential smoothing, ARIMA, tree-based models, and linear regression (Zeng et al., 2023), as well as most neural networks including LSTNet (Lai et al., 2018), DSANet (Huang et al., 2019), TPA (Shih et al., 2019), Leddam (Yu et al., 2024), and iTransformer (Liu et al., 2024). The autoregressive assumption enables these models to capture periodicities quite effectively without specialized modules, but is somewhat constrained by the autoregressive formulation. Essentially, periodicity implies that the data-generating process (DGP) involves $t$ (i.e., $x_t = f(t, x_{t-1}, x_{t-2}, \dots)$), but $t$ is not an input in an autoregressive process.

Explicit methods employ specialized components to directly capture periodicity and improve model performance. For example, one can use Fourier transforms or positional embeddings to encode periodicity information. However, this introduces additional input to the model, so the model now relies on auxiliary inputs (i.e., exogenous variables) instead of depending solely on historical values (i.e., endogenous variables). Thus, strictly speaking, such exogenous models are not comparable to endogenous ones. Our paper focuses on a model agnostic method for improving neural networks' per-

formance on long-term periodic time series forecasting by explicitly capturing periodicity through alignment without auxiliary input.

There are many previous methods such as ETS (error, trend, seasonal) and Seasonal ARIMA (SARIMA) that are developed based on traditional models and are not combinable with neural networks. In terms of neural networks, most previous methods, such as SparseTSF (Lin et al., 2024b), DEPTS (Fan et al., 2022), and Leddam (Yu et al., 2024), directly infuse periodic information into the middle of the model through variations of seasonal-trend decomposition. However, SFNN (Sun et al., 2025) has shown that a simple feed-forward architecture can explicitly capture periodicity and achieves the best performance. Other than directly infusing periodic information, CycleNet (Lin et al., 2024a) is the most comparable method to ours because their approach is also combinable with any neural network and does not rely on auxiliary input. Therefore, we compare against them in the main experiment.

In this work, we propose PAPER: **P**eriodicity **A**lignment on **Per**iodic time series for forecasting. This alignment mechanism is a simple yet effective approach without auxiliary input that enhances the model's ability to capture periodic patterns while enabling the learning of non-autoregressive temporal dependencies.

Our key contributions are as follows.

- We introduce PAPER, a periodicity alignment framework for periodic time series that improves state-of-the-art model performance by learning non-autoregressive dependencies across temporal contexts without auxiliary input.

- We provide theoretical analysis with mathematical proofs that characterize both the advantages and fundamental limitations of periodicity alignment.

- We demonstrate the properties and effectiveness of our alignment approach through controlled experiments on synthetic datasets with known periodic structures.

- We conduct comprehensive experiments on real-world datasets that validate the practical benefits of our method and clearly delineate its scope and limitations.

## 2 PROBLEM FORMULATION

### 2.1 OUR DEFINITION OF ALMOST PERIODIC TIME SERIES

Let $\mathbf{x} = \{x_1, \ldots, x_T\}$ be a univariate time series of length $T$. We say $\mathbf{x}$ is periodic with fundamental period $P$ when $x_{t+P} = x_t$ for all $t$. However, perfectly periodic time series are rare in the real world. Instead, we focus on time series that are almost periodic. Our definition of an almost periodic signal with a *fixed* fundamental period $P$ is:

$$\mathbb{E}(x_{t+P} - x_t)^2 \leq \epsilon^2, \forall t, \tag{1}$$

where $\epsilon$ is a small noise term. This is a simplified definition of the almost periodic function of Bohr (1925), which is also similar to quasi-periodic motion (in mathematics and theoretical physics) or quasi-periodic signals (in signal processing). Compared to the almost periodic function in Bohr (1925), the $P$ and $\epsilon$ in our definition are the result of the nature of the data-generating process, rather than arbitrary choices. Compared to quasiperiodicity, our definition employs a fixed fundamental period $P$, instead of a possibly slowly changing $P$.

### 2.2 TIME SERIES FORECASTING

In time series forecasting, we aim to forecast the future $H$ horizons $[x_{t+1}, \ldots, x_{t+H}]$, given the past histories $[x_1, \ldots, x_t]$. Thus, this is a sequence-to-sequence task where a single example is of the form (target, input) $= ([x_{t+1}, \ldots, x_{t+H}], [x_1, \ldots, x_t])$. Since more recent histories are more informative, a common approach is to select a suitable look-back length $L$ of the most recent histories as the input to the model. When the time series is $N$-multivariate, we assume the $N$ series start at the same time, have the same length of $T$, have the same sampling rate, and most importantly, have the same fundamental period $P$. We use $\mathbf{X} = [\mathbf{X}_1, \mathbf{X}_2, \ldots, \mathbf{X}_T]^\top \in \mathbb{R}^{T \times N}$ to denote the $N$-multivariate time series, where $\mathbf{X}_t \in \mathbb{R}^N$ is the $t$-th sample for all series. The (target, input)

pair in the multivariate case is the same as the univariate case but with $\mathbf{X}_t$ as each sample. Ultimately, we aim to find the model parameters $f(\cdot;\theta)$ that minimize the mean squared error (MSE) $= \sum_t \sum_{h=1}^{H} \|\mathbf{X}_{t+h} - \hat{\mathbf{X}}_{t+h}\|_2^2$ on the test set, where $[\hat{\mathbf{X}}_{t+1},\ldots,\hat{\mathbf{X}}_{t+H}]^\top = f(\mathbf{X}_t,\ldots,\mathbf{X}_{t-L+1};\theta)$ is the model forecast.

## 3 OUR METHOD

As indicated in the title of this paper, our method assumes that the time series of interest has strong fundamental periodicity. Thus, our method consists of two steps. First, a forecasting-based method to detect the fundamental periodicity. Second, periodicity alignment is performed to encourage the model to make use of periodic patterns.

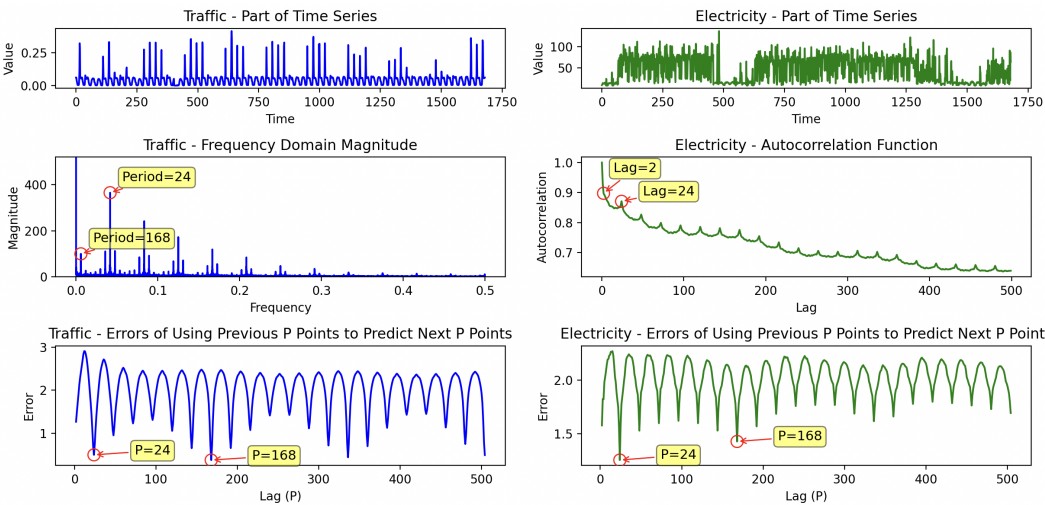

Figure 1: Three views to detect the best periodicity in the Traffic and Electricity datasets. The top row shows the original time series where one can visually understand the periodicity. The middle row shows results using Fourier transform (for Traffic) or autocorrelation function (for Electricity). The bottom row shows results using our method of periodicity detection. On the Traffic dataset, Fourier transform failed to detect the weekly periodicity as the best one because sinusoidal bases are not suitable. On the Electricity dataset, the changing trends make it difficult to identify 24 as the best periodicity. In contrast, our method successfully detects the best periodicities in both cases while being simple and straightforward.

### 3.1 DETECTING THE FUNDAMENTAL PERIODICITY

Traditionally, there are two main methods to detect periodicity in time series: Fourier (or wavelet) transform or autocorrelation function. Advanced methods are built on top of these (Puech et al., 2019; Wen et al., 2021). For example, Wen et al. (2021) uses the wavelet transform first and then applies the autocorrelation function to improve robustness. Instead of relying on more sophisticated methods, we employ a simple method that best suits our use case of forecasting. Essentially, we directly measure the error of using the previous $P$ data points to predict the next $P$ data points and find the $P$ that minimizes the error. Mathematically, the best fundamental period $P^\star$ is found according to this equation:

$$P^\star = \arg\min_P \frac{1}{|\mathcal{T}|} \sum_{t\in\mathcal{T}} \|\mathbf{X}_{t-P+1:t} - \mathbf{X}_{t+1:t+P}\|_F^2, \qquad (2)$$

where $\mathbf{X}_{i:j} = [\mathbf{X}_i \ \ldots \ \mathbf{X}_j]^\top$ and $\mathcal{T}$ is the set of time index. To further increase robustness to trend and scale drift, each of the $N$ series in $\mathbf{X}_{t-P+1:t}$ and $\mathbf{X}_{t+1:t+P}$ is $z$-normalized individually before the Frobenius norm is calculated. One can immediately understand why detecting periodicity using our method is very suitable when the use case is forecasting. Qualitatively speaking, using Fourier

(or wavelet) transforms is sometimes suboptimal because the shapes can be very different from sinusoidal bases, and using autocorrelation functions is sensitive to trends. Our method alleviates both problems. As an illustration, Figure 1 visualizes the advantages of our method.

## 3.2 PERIODICITY ALIGNMENT

After finding the best periodicity, we align the series according to its periodicity so that the model knows exactly which input corresponds to which step in the period. For example, assuming we are dealing with daily data points with weekly periodicity, then after alignment, the first position always corresponds to Monday. This differs from typical time series forecasting, where the input consists of rolling slices of the whole time series with the temporal order preserved. This means that the starting time step of the input can be any day in the week. For a visual comparison, see Figure 2.

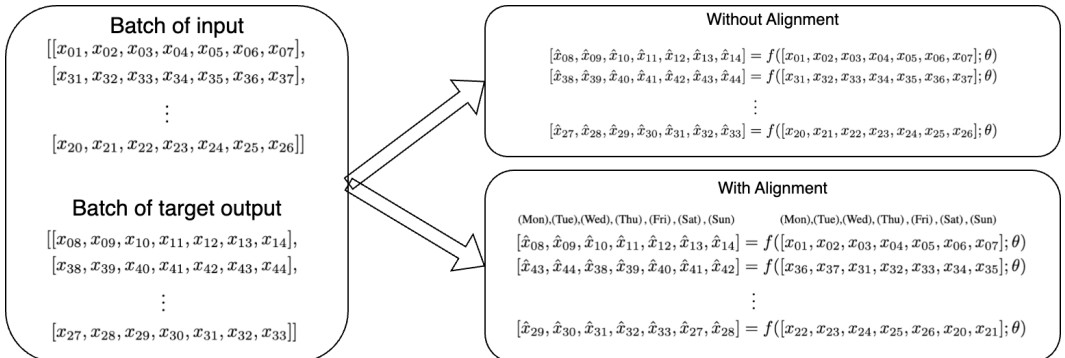

Figure 2: An illustration of the difference between with or without periodicity alignment. In the left block, we are given a batch of randomly shuffled input and also the target output. Without alignment, the input and output are unchanged but then the periodic patterns are shifting over time. In contrast, with alignment, the input and output are wrapped so that the each position always corresponds to its weekday. This way the periodic pattern is not shifting but the entries are not temporally ordered.

### 3.2.1 ADDING POSITIONAL INFORMATION

An apparent issue with such periodicity alignment is the problem of temporally unordered entries. This is problematic because in most cases, the nearest data point $x_t$ has the most predictive power, but now the model does not know which position corresponds to $x_t$. Our solution is to expand the input vector instead of wrapping the series during alignment, so now the input vector length is $L + P - 1$. Following the batch of input in Figure 2, with expansion, the input becomes

$$\text{(Mon),(Tue),(Wed), (Thu) , (Fri) , (Sat) , (Sun),(Mon),(Tue),(Wed), (Thu) , (Fri) , (Sat)}$$
$$f([x_{01}, x_{02}, x_{03}, x_{04}, x_{05}, x_{06}, x_{07}, \quad 0, \quad 0, \quad 0, \quad 0, \quad 0, \quad 0]; \theta)$$
$$f([\quad 0, \quad 0, x_{31}, x_{32}, x_{33}, x_{34}, x_{35}, x_{36}, x_{37}, \quad 0, \quad 0, \quad 0, \quad 0]; \theta)$$
$$\vdots$$
$$f([\quad 0, \quad 0, \quad 0, \quad 0, \quad 0, x_{20}, x_{21}, x_{22}, x_{23}, x_{24}, x_{25}, x_{26}, \quad 0]; \theta).$$

The same expansion also applies to the output. An ablation study in Section 5.4 indeed confirms that such expansion is beneficial to model performance.

## 4 ANALYSES OF PAPER ALIGNMENT

In this section, we conduct a comprehensive analysis of PAPER alignment from multiple perspectives. We begin with a qualitative examination of the fundamental advantages and limitations of PAPER alignment. We then provide theoretical foundations for these observations through rigorous mathematical analysis and formal proofs. Subsequently, we validate our theoretical findings using controlled experiments on two synthetic datasets designed to isolate specific properties of the alignment mechanism. Finally, we demonstrate the superiority of alignment as a dimensionality reduction technique by comparing its efficiency against established methods such as the Fourier transform.

## 4.1 QUALITATIVE ANALYSES

By explicitly aligning periodicity, the data are in some sense simplified from the model's perspective. A simple illustration is shown in Figure 3, where one can immediately see the difference between with and without alignment. Without alignment, the input data are messier; with alignment, the input data are cleaner. Mathematically speaking, this implies that the input data have lower rank after alignment. Thus, the model converges faster and similar performance can potentially be achieved with a much smaller model, which is proven in the next section under some reasonable assumptions.

Another observation is that, without alignment, the model assumes that the data-generating process (DGP) is strictly autoregressive. For example, this means that in traffic forecasting, the model remains the same regardless of whether the prediction is for Monday or Saturday. In contrast, with alignment, we are essentially training $P$ different models where the parameters are shared so that the total parameter size remains the same. A toy example in Section 4.4 shows the benefits of alignment when the data are not autoregressive.

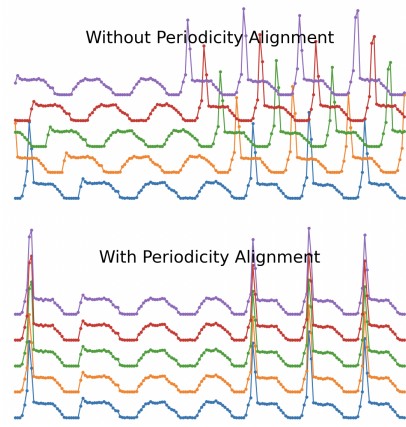

Figure 3: A simple visualization of what happens to the input data without alignment versus with alignment on the Traffic dataset. Qualitatively speaking, the input data with alignment is cleaner.

## 4.2 MATHEMATICAL PROPERTIES OF PERIODICITY ALIGNMENT

To simplify and derive several important mathematical properties of alignment, we make some reasonable assumptions about the model and the data in Assumption 1.

**Assumption 1.** *Following the symbols used in Section 2, the assumptions are:*

- *The data-generating process (DGP) is $x_{t+P} = x_t + e_t$, where $P \geq 2$, $e_t \sim \mathcal{N}(0, \epsilon^2)$ is i.i.d., and the values of $\{x_{1-P}, \ldots, x_0\}$ are some fixed non-trivial constants. This ensures that the time series is almost periodic as introduced in Section 2.1.*

- *The model of interest is a linear model.*

- *The length of the time series is much greater than its period (i.e., $T \gg P$).*

- *The look-back length and the horizon are both equal to the period (i.e., $L = H = P$).*

*Collectively, these assumptions are referred to as Assumption 1.*

Under Assumption 1, we can derive Theorem 4.1.

**Theorem 4.1.** *Under Assumption 1, applying periodicity alignment decreases the training error. Specifically, the residual sum of squares (RSS) without alignment is $(T - P)P\epsilon^2$, and the RSS with alignment is at most $(T - 2P + 1)P\epsilon^2$, which is strictly less than the RSS without alignment.*

*Proof.* Please see Section A.1 for the proof. □

An interesting observation based on Theorem 4.1 is that with exactly the same model and data, aligning the periodicity strictly improves the training error. Although this is promising, we note that the testing error actually increases with alignment, as stated in Theorem 4.2.

**Theorem 4.2.** *Under Assumption 1, applying periodicity alignment increases the testing error. Specifically, the testing error without alignment is $P(1 + \frac{P}{T})\epsilon^2$, and the testing error with alignment is at least $P(1 + \frac{2P+1}{T})\epsilon^2$, which is strictly greater than the testing error without alignment.*

*Proof.* Please see Section A.2 for the proof. □

This is certainly a limitation of periodicity alignment. However, this is due to the fact that the DGP in Assumption 1 is autoregressive, so there is no need for alignment since a linear autoregressive model is already the best estimator. That said, alignment still has an advantage under reduced-rank regression (RRR), where the rank of the model is restricted, as shown in Theorem 4.3.

**Theorem 4.3.** *Under Assumption 1, we already know that applying periodicity alignment decreases the training error according to Theorem 4.1. Furthermore, if the rank of the model is restricted, the increase in training error is smaller when alignment is applied, implying that the outperformance of alignment is magnified. Specifically, the increase in residual sum of squares (RSS) with rank $r < P$ is roughly $(P - r)(\frac{T||\mathbf{x}_0||_2^2}{P} + \frac{T(T+P)\epsilon^2}{2P})$ without alignment and $T||\mathbf{x}_0||_2^2 + (P - r)\frac{T(T+P)\epsilon^2}{2P}$ with alignment.*

*Proof.* Please see Section A.3 for the proof. $\square$

Comparing the test error of RRR models with and without alignment is subtle because two opposing forces are at play. As shown in Theorem 4.2, alignment tends to raise test error. In contrast, Theorem 4.3 shows that when the model rank is reduced, alignment improves training performance. The overall outcome depends on which influence prevails: if the rank is trimmed only slightly, alignment may degrade performance, but when the rank is cut substantially, alignment can yield better results.

### 4.3 Autoregressive and Non-Autoregressive Toy Examples

In this section, we demonstrate an experimental proof of Theorem 4.1 and Theorem 4.2. Following Assumption 1, we construct a toy dataset of various sample sizes $T = \{200, 666, 2000, \ldots, 200000\}$ with an $80\%/20\%$ train-test split and set $P = H = 5, \epsilon = 1$. Additionally, we ran 100 runs for each setup, and for each run, we randomly sampled the values $\{x_{-4}, x_{-3}, \ldots, x_0\}$ from $\mathcal{N}(0, 10^2)$. The results are shown in Figure 4a. When $T$ is large, it does not matter whether alignment is applied or not. However, when $T$ is not sufficiently large, we can see that the training error is lower while the testing error is higher after alignment. This follows Theorem 4.1 and Theorem 4.2. We also show the estimated errors according to the equations in the proofs.

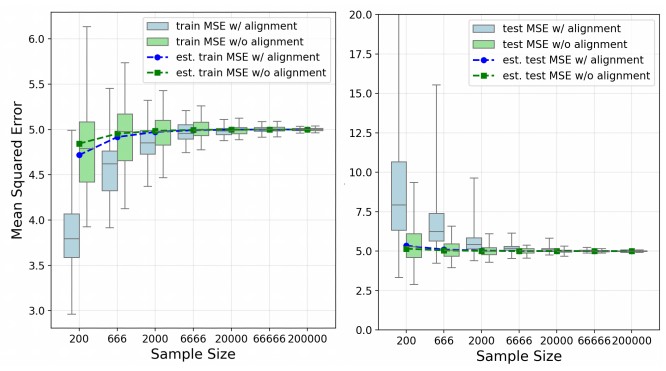
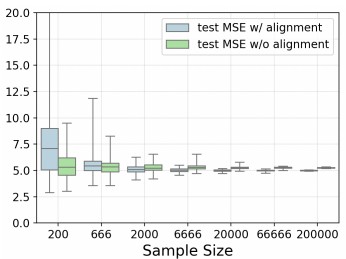

(b) Here, the synthetic toy datasets are generated similarly to Figure 4a, but are no longer autoregressive. Instead, we have $x_t = a_t x_{t-P} + e_t$, where $a_t \in \{0.8, 0.85, 0.9, 0.95, 1\}$ and is selected based on $(t \bmod P)$. For example, if $(t \bmod P) = 3$, then $a_t = 0.95$.

(a) The synthetic toy datasets are autoregressively generated according to Assumption 1. We set an $80\%/20\%$ train-test split and $P = H = 5, \epsilon = 1$. Each quartile box plot is calculated over 100 runs.

Figure 4: Affect of applying alignment on synthetic toy datasets.

### 4.4 Non-Autoregressive Examples

After seeing the results in the previous section, one might question the benefits of PAPER alignment. However, notice that the data-generating process in Assumption 1 is exactly autoregressive and therefore exactly matches the formulation without alignment. On the other hand, the alignment formulation is closer to training $P$ different models with shared parameters, which is less efficient when the data are autoregressive. This is what we described in the last paragraph of our qualitative analysis in Section 4.1. An example of a non-autoregressive DGP is when the parameters are dependent on the phase in the period. For example, in traffic forecasting, the best model for predicting

Monday's traffic flow will be different from the best model for Saturday's. This is because, first, Monday is a weekday and Saturday is a weekend, causing different human behaviors. Additionally, the day before Monday is a weekend, whereas the day before Saturday is a weekday, so if the model assigns the same coefficient to the previous day, the performance is suboptimal.

Since it is difficult to prove this mathematically, we construct a non-autoregressive toy dataset to demonstrate it in the following. The non-autoregressive toy datasets follow the process $x_t = a_t x_{t-P} + e_t$, where $a_t \in \{0.8, 0.85, 0.9, 0.95, 1\}$ and is selected based on $(t \mod P)$. For example, if $(t \mod P) = 1$, then $a_t = 0.85$, and if $(t \mod P) = 3$, then $a_t = 0.95$. Otherwise, the process is the same as in the previous section. The results on the training set are similar to those of Figure 4a, where the error is again much lower. The difference lies in the results of the test set, which are shown in Figure 4b. We can clearly see that applying alignment now improves performance with sufficient sample size. Moreover, the improvement remains even with larger sample sizes.

Additionally, we perform a proof-of-concept experiment on real-world datasets. Specifically, we split the data into $P$ subsets according to the phase and then train $P$ linear models. The results show significant improvements compared to using a single linear model. On the Electricity, Solar, and Traffic datasets with $P = 24, 144, 168$, the improvements are 45%, 22%, and 13%, respectively.

### 4.5 Dimensionality Reduction Analysis

In this analysis, we aim to show that PAPER alignment helps with dimensionality reduction. This means that after alignment, the data matrix $\mathbf{X}$ can be projected to a low-dimensional space with less distortion, and thus achieve lower reconstruction error when projecting back to the original high-dimensional space. The results are shown in Figure 5, where one can see that simply aligning periodicity before Principal Component Analysis (PCA) achieves the lowest reconstruction error when compared to PCA only, kernel PCA, Discrete Fourier Transform (DFT), and Wavelet Transform. Outperforming PCA alone is straightforward by looking at Figure 3 visually, but the fact that PAPER also outperforms kernel PCA, DFT, and Wavelet Transform implies that PAPER is an extremely efficient representation of the data.

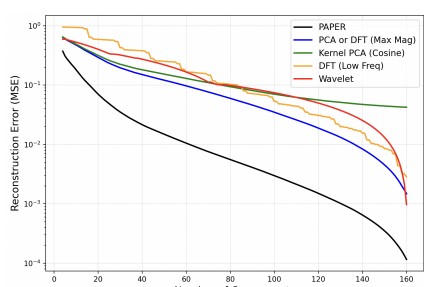

Figure 5: Reconstruction error on the first series in the Traffic dataset using dimensionality reduction techniques. The series is transformed to a 2D matrix where each row is a sliding window of length 168. With PAPER alignment, the rows are aligned before passing to PCA. PCA and DFT with maximum magnitude achieve extremely similar results due to both being linear operations.

## 5 Real-World Experiments

In this section, we perform several experiments on real-world datasets that solidify the conclusions drawn from Section 4.3, Section 4.4, and Section 4.5.

### 5.1 Long-term Forecasting Results

Our experimental setup is similar to previous long-term forecasting papers in the deep learning community, except for the train-validation-test split. Previous work mostly follows a chronologically ordered train-validation-test split, so the best validation epoch is tested on the test set. However, this deviates from real-world practice because the validation set is closest to the test set, so not training on it is too costly. Instead, practitioners usually first find suitable hyperparameters that do not overfit and then train the model on the full available training set. Additionally, our PAPER alignment is prone to overfitting as proven in Theorem 4.2 and further discussed in Section 6.2, so a validation set between the training set and test set greatly reduces performance. In summary, we set the first 95% of the data as the training set and the remaining 5% as the test set, and choose the best hyperparameters for each model to train for a fixed number of epochs before testing. For ease of comparison, the models are identical, where the number of units is 336, 144, 1344 for the Electricity dataset, Solar dataset, and Traffic dataset, respectively. Thus, the tunable hyperparameters pertain to the optimization process, not the model itself.

Table 1: Mean-Squared Errors (MSEs) on the test set with various model combinations for three datasets and four horizons. Mean and standard deviation reported over 10 runs. Shaded number indicates the best performing model and is superscribed with † if the outperformance is statistically significant with $p$-value less than $5\%$.

| Dataset | Horizon | SFNN | SFNN + CycleNet | SFNN + PAPER |
|---|---|---|---|---|
| Electricity | 168 | $0.1646 \pm 0.0012$ | $0.1622 \pm 0.0002$ | $0.1589^\dagger \pm 0.0002$ |
| | 336 | $0.1661 \pm 0.0006$ | $0.1623 \pm 0.0003$ | $0.1591^\dagger \pm 0.0002$ |
| | 504 | $0.1813 \pm 0.0003$ | $0.1766 \pm 0.0004$ | $0.1674^\dagger \pm 0.0004$ |
| | 672 | $0.1985 \pm 0.0003$ | $0.1975 \pm 0.0004$ | $0.1836^\dagger \pm 0.0003$ |
| Solar | 144 | $0.1889^\dagger \pm 0.0020$ | $0.2188 \pm 0.0003$ | $0.2047 \pm 0.0052$ |
| | 288 | $0.2125 \pm 0.0024$ | $0.2675 \pm 0.0007$ | $0.2020^\dagger \pm 0.0028$ |
| | 432 | $0.2212 \pm 0.0005$ | $0.2742 \pm 0.0003$ | $0.2090^\dagger \pm 0.0015$ |
| | 576 | $0.2244 \pm 0.0007$ | $0.2620 \pm 0.0007$ | $0.2239 \pm 0.0015$ |
| Traffic | 168 | $0.3314 \pm 0.0002$ | $0.3329 \pm 0.0012$ | $0.3319 \pm 0.0015$ |
| | 336 | $0.3512 \pm 0.0008$ | $0.3457 \pm 0.0009$ | $0.3384^\dagger \pm 0.0009$ |
| | 504 | $0.3557 \pm 0.0001$ | $0.3590 \pm 0.0013$ | $0.3430^\dagger \pm 0.0001$ |
| | 672 | $0.3761 \pm 0.0004$ | $0.3910 \pm 0.0021$ | $0.3589^\dagger \pm 0.0007$ |

The results are shown in Table 1. We choose SFNN (Sun et al., 2025) as the base model because it is the state-of-the-art model, yet it has a very simple architecture that does not interfere with our alignment. Compared to SFNN without alignment or SFNN with CycleNet, adding PAPER alignment improves performance in most cases, with improvement up to 7%. Notice that the improvement seems to decrease as the horizon shortens, which we will discuss more in Section 6.3.

## 5.2 AGNOSTIC TO DIFFERENT MODELS

In the previous experiment, we chose SFNN (Sun et al., 2025) as the base model. Here, we show that PAPER alignment is model-agnostic, so it can also work with other base models, such as Leddam (Yu et al., 2024), NLinear (Zeng et al., 2023), and iTransformer (Liu et al., 2024). From the results in Figure 6, we can see that adding PAPER alignment improves performance for all models across most datasets and horizons.

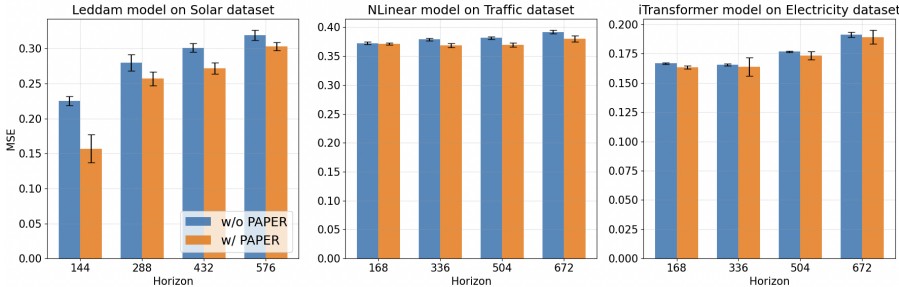

Figure 6: MSEs on the test set for various base models on various datasets and horizons. The error bars indicate 2 standard deviations over 10 runs.

## 5.3 REDUCING MODEL SIZES

As described in Theorem 4.3 and Section 4.5, using PAPER helps when we want a smaller model. To demonstrate this property of PAPER, we train SFNNs with increasingly fewer hidden units. The results on the Traffic dataset with horizon 168 are shown in Figure 7. We can see that PAPER alignment greatly improves performance when the number of hidden units is reduced. Specifically, comparing 21 versus 1344 hidden units, the model size is just 0.5% but the error only increases by 11% (instead of around 26.5% with CycleNet or without alignment).

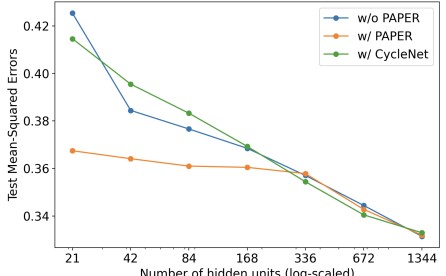

Figure 7: MSEs on the Traffic dataset with horizon 168 using various numbers of hidden units. The results show that with PAPER, the degradation of performance under restricted model sizes is much less severe. This follows Theorem 4.3 and matches Figure 5.

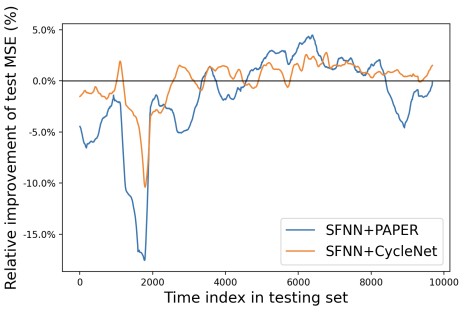
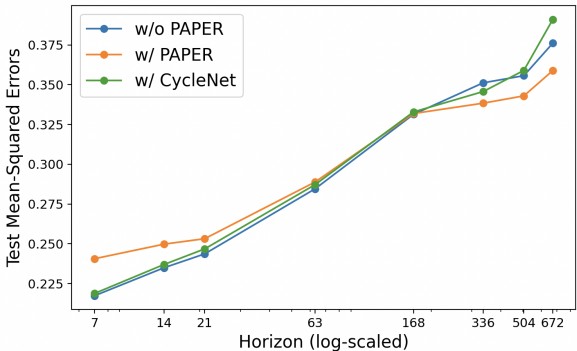

(a) The figure shows the relative improvement of SFNN after applying PAPER or CycleNet. The models are trained on 60% of the Electricity dataset with horizon 168. Since our alignment method is more prone to concept drift in the data as proven in Theorem 4.2, the improvement decreases faster over time. Additionally, the improvement is more volatile with PAPER.

(b) MSEs on the Traffic dataset with various target horizons from short- to long-term, while the look-back length is fixed at 1344. When the horizon is short, applying PAPER negatively impacts performance. However, when the horizon is longer, especially when it is at least one period long (168 in this case), PAPER greatly improves performance.

## 5.4 ABLATION STUDY

In Figure 2, the original PAPER alignment does not include positional information, which we call rolling PAPER. An improvement we made is described in Section 3.2.1, where we add positional information to PAPER alignment by expanding the input size by $P - 1$. An ablation study tabulated in Table 2 in Appendix confirms the addition of positional information is beneficial to performance.

## 6 LIMITATIONS

Our PAPER alignment is a simple method to enhance performance for periodic time series. However, there are still three major limitations of which one should be aware.

### 6.1 LESS IMPROVEMENT UNDER MULTIPLE PERIODICITY

Our method's core functionality relies on the existence of a single, strong fundamental periodicity within the time series data. Consequently, its efficacy is significantly diminished or eliminated when applied to non-periodic time series. Furthermore, even in cases where multiple fundamental periodicities are present, the overall improvement provided by our approach is reduced. For instance, the experimental results detailed in Table 1 (referencing ETTm1 and ETTh1 datasets) demonstrate that the improvement becomes significant only when the prediction horizon is long.

### 6.2 MORE SENSITIVE TO DISTRIBUTION DRIFT

First, as proven in Theorem 4.2, PAPER alignment is more prone to overfitting. Thus, a slight change in the pattern can have a greater effect on its performance. To demonstrate this limitation, we train SFNN with PAPER alignment on 60% of the Electricity dataset to predict the future 168 horizons and then observe how the model performs on the remaining 40% of the data. We also compare against CycleNet. The results are shown in Figure 8a, where we can observe two things: (1) the improvement of PAPER is more concentrated at the beginning of the test set but then deteriorates quite rapidly; and (2) the improvement of PAPER is more volatile when compared to CycleNet. To mitigate distribution drift, we use stronger L2 regularization and dropout.

### 6.3 DEGRADATION ON SHORT-TERM FORECASTING

The second limitation is that PAPER alignment does not work well on short-term forecasting, especially when the horizon is shorter than the period. An experiment predicting short horizons confirms this limitation. As shown in Figure 8b, alignment actually harms performance when the prediction horizon is less than the period. On the other hand, the benefits of PAPER alignment start to emerge when the horizon is longer than the period and become increasingly beneficial for longer horizons.

## 7 CONCLUSION

In conclusion, we introduced PAPER, a novel periodicity alignment framework for long-term periodic time series forecasting. Our approach further improves state-of-the-art model performance by learning non-autoregressive dependencies without auxiliary input. We utilized theoretical proofs, synthetic datasets, and real-world experiments to confirm the practical benefits of our method while also clearly defining its limitations. Our work provides an initial exploration of periodicity alignment that paves the way for further research on more sophisticated methods.

## REPRODUCIBILITY STATEMENT

The source code, configuration files, and instructions for model training will be provided for all key experiments upon acceptance. Proofs, with their underlying assumptions and explanations, are detailed in the paper and the appendix.

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

# Appendices

## A    MATHEMATICAL PROPERTIES OF PERIODICITY ALIGNMENT

Under Assumption 1, the fitting process is essentially a multi-target multiple linear regression:

$$\mathbf{Y} = \mathbf{X}\mathbf{B} + \mathbf{E}, \tag{3}$$

where $\mathbf{Y} \in \mathbb{R}^{T \times P}$ is the target matrix, $\mathbf{X} \in \mathbb{R}^{T \times P}$ is the input matrix, $\mathbf{E} \in \mathbb{R}^{T \times P}$ is the noise matrix, and $\mathbf{B} \in \mathbb{R}^{P \times P}$ is the parameter matrix to be learned. For simplicity, here the whole time series has length of $T + 2P - 1$ (instead of $T$).

Without alignment, we have

$$\mathbf{X} = \begin{bmatrix} x_1 & x_2 & \ldots & x_P \\ x_2 & x_3 & \ldots & x_{P+1} \\ \vdots & \vdots & \ddots & \vdots \\ x_T & x_{T+1} & \ldots & x_{T+P} \end{bmatrix} \in \mathbb{R}^{T \times P}, \mathbf{Y} = \begin{bmatrix} x_{P+1} & x_{P+2} & \ldots & x_{2P} \\ x_{P+2} & x_{P+3} & \ldots & x_{2P+1} \\ \vdots & \vdots & \ddots & \vdots \\ x_{T+P} & x_{T+P+1} & \ldots & x_{T+2P-1} \end{bmatrix} \in \mathbb{R}^{T \times P}, \tag{4}$$

whereas with alignment, we have

$$\mathbf{X} = \begin{bmatrix} x_1 & x_2 & \ldots & x_P \\ x_{P+1} & x_2 & \ldots & x_P \\ x_{P+1} & x_{P+2} & \ldots & x_P \\ \vdots & \vdots & \ddots & \vdots \end{bmatrix} \in \mathbb{R}^{T \times P}, \mathbf{Y} = \begin{bmatrix} x_{P+1} & x_{P+2} & \ldots & x_{2P} \\ x_{2P+1} & x_{P+2} & \ldots & x_{2P} \\ x_{2P+1} & x_{2P+2} & \ldots & x_{2P} \\ \vdots & \vdots & \ddots & \vdots \end{bmatrix} \in \mathbb{R}^{T \times P}. \tag{5}$$

Keep in mind that under the assumptions, $\hat{\mathbf{B}} = \mathbf{I}$ is the optimal solution for testing set (but not for training set) and $\mathbf{Y} = \mathbf{X} + \mathbf{E}$.

### A.1    REDUCTION IN TRAINING ERROR

Here, we show that with alignment, the training error is always lower. Notice that this is achieved with the same model and data. The only difference is in how the data is fed into the model.

According to the multi-target multiple linear regression, the solution of $\mathbf{B}$ is

$$\hat{\mathbf{B}} = (\mathbf{X}^\top \mathbf{X})^{-1} \mathbf{X}^\top \mathbf{Y} \tag{6}$$

$$= (\mathbf{X}^\top \mathbf{X})^{-1} \mathbf{X}^\top (\mathbf{X} + \mathbf{E}) \tag{7}$$

$$= \mathbf{I} + (\mathbf{X}^\top \mathbf{X})^{-1} \mathbf{X}^\top \mathbf{E}. \tag{8}$$

Here, we can already see that $\hat{\mathbf{B}}$ is indeed close to an identity matrix but with some perturbations due to the noise. The residual sum of squares (RSS) is thus

$$\text{RSS} = \|\mathbf{Y} - \mathbf{X}\hat{\mathbf{B}}\|_F^2 \tag{9}$$

$$= \|\mathbf{X} + \mathbf{E} - \mathbf{X}(\mathbf{I} + (\mathbf{X}^\top\mathbf{X})^{-1}\mathbf{X}^\top\mathbf{E})\|_F^2 \tag{10}$$

$$= \|\mathbf{E} - \mathbf{X}(\mathbf{X}^\top\mathbf{X})^{-1}\mathbf{X}^\top\mathbf{E}\|_F^2 \tag{11}$$

$$\coloneqq \|(\mathbf{I} - \mathbf{P})\mathbf{E}\|_F^2. \tag{12}$$

where $\|\cdot\|_F^2$ is the Frobenius norm and $\mathbf{P} = \mathbf{X}(\mathbf{X}^\top\mathbf{X})^{-1}\mathbf{X}^\top$ is the projection matrix. Additionally, projection matrix has the following properties: (1) $\mathbf{P} = \mathbf{P}^\top = \mathbf{P}^2$, and (2) $\mathbf{P} = \mathbf{U}\mathbf{U}^\top$, where $\mathbf{U}$ are the first $P$ left singular vectors of $\mathbf{X}$. Using these properties, we can further simplify Equation (12):

$$\text{RSS} = \|(\mathbf{I} - \mathbf{P})\mathbf{E}\|_F^2 \tag{13}$$

$$= \text{tr}(\mathbf{E}^\top(\mathbf{I} - \mathbf{P})(\mathbf{I} - \mathbf{P})\mathbf{E}) \tag{14}$$

$$= \text{tr}(\mathbf{E}\mathbf{E}^\top(\mathbf{I} - \mathbf{P})) \tag{15}$$

$$= \text{tr}(\mathbf{E}\mathbf{E}^\top) - \text{tr}(\mathbf{E}\mathbf{E}^\top\mathbf{P}) \tag{16}$$

$$= \|\mathbf{E}\|_F^2 - \text{tr}(\mathbf{E}\mathbf{E}^\top\mathbf{U}\mathbf{U}^\top) \tag{17}$$

$$= \|\mathbf{E}\|_F^2 - \|\mathbf{E}^\top\mathbf{U}\|_F^2, \tag{18}$$

where $\|\mathbf{E}\|_F^2 \approx TP\epsilon^2$ is the same with or without alignment.

### A.1.1 RSS WITHOUT ALIGNMENT

Without alignment, we have

$$\mathbf{E} = \begin{bmatrix} e_1 & e_2 & \ldots & e_P \\ e_2 & e_3 & \ldots & e_{P+1} \\ \vdots & \vdots & \ddots & \vdots \\ e_T & e_{T+1} & \ldots & e_{T+P} \end{bmatrix} \in \mathbb{R}^{T \times P}. \tag{19}$$

Following Equation (18), we want to estimate the value of $\|\mathbf{E}^\top\mathbf{U}\|_F^2$ by examining each elements in the matrix:

$$(\mathbf{E}^\top\mathbf{U})_{ij}^2 = (\sum_{t=1}^{T} \mathbf{E}_{ti}\mathbf{U}_{tj})^2 \tag{20}$$

$$= \sum_{t=1}^{T} \mathbf{E}_{ti}^2\mathbf{U}_{tj}^2 + \sum_{t=1}^{T} \sum_{s=1,s\neq t}^{T} \mathbf{E}_{ti}\mathbf{E}_{si}\mathbf{U}_{tj}\mathbf{U}_{sj}. \tag{21}$$

Then,

$$\|\mathbf{E}^\top\mathbf{U}\|_F^2 = \sum_{i=1}^{P}\sum_{j=1}^{P}(\mathbf{E}^\top\mathbf{U})_{ij}^2 \tag{22}$$

$$= \sum_{i=1}^{P}\sum_{j=1}^{P}\left[\sum_{t=1}^{T}\mathbf{E}_{ti}^2\mathbf{U}_{tj}^2 + \sum_{t=1}^{T}\sum_{s=1,s\neq t}^{T}\mathbf{E}_{ti}\mathbf{E}_{si}\mathbf{U}_{tj}\mathbf{U}_{sj}\right] \tag{23}$$

$$= \sum_{j=1}^{P}\sum_{t=1}^{T}\left[\mathbf{U}_{tj}^2\sum_{i=1}^{P}\mathbf{E}_{ti}^2\right] + \sum_{i=1}^{P}\sum_{j=1}^{P}\sum_{t=1}^{T}\sum_{s=1,s\neq t}^{T}\mathbf{E}_{ti}\mathbf{E}_{si}\mathbf{U}_{tj}\mathbf{U}_{sj}. \tag{24}$$

$$\tag{25}$$

In the first part of the summation, we can see that $\forall t, \sum_{i=1}^{P}\mathbf{E}_{ti}^2$ are all the same distribution and have mean of $P\epsilon^2$. Thus, the first term roughly sums up to $P^2\epsilon^2$. The second part of the summation roughly sums up to $0$ because $\mathbf{E}_{ti}$ and $\mathbf{E}_{si}$ are two independent Gaussian distributions (i.e., $\mathbb{E}[\mathbf{E}_{ti}\mathbf{E}_{si}] = 0$) and they are (almost) independent of $\mathbf{U}$. Thus, the RSS without alignment is roughly

$$\text{RSS}_{\text{w/o}} \approx TP\epsilon^2 - P^2\epsilon^2 = (T - P)P\epsilon^2. \tag{26}$$

### A.1.2 RSS WITH ALIGNMENT

Again, we start from Equation (18). However, this time, $\mathbf{E}$ is structurally different:

$$\mathbf{E} = \begin{bmatrix} e_1 & e_2 & \ldots & e_P \\ e_{P+1} & e_2 & \ldots & e_P \\ e_{P+1} & e_{P+2} & \ldots & e_P \\ \vdots & \vdots & \ddots & \vdots \end{bmatrix} \in \mathbb{R}^{T \times P}. \tag{27}$$

Following similar derivations, we can arrive at the same Equation (24). The first part of the summation has the same distribution with or without alignment, but the second part is different. The reason is that $\mathbf{E}_{ti}$ and $\mathbf{E}_{si}$ are no longer two independent Gaussian distributions because of the alignment. For example, in the first column in $\mathbf{E}$, $e_{P+1}$ appears $P$ times after alignment (see Equation (27)), but only appears one time without alignment (see Equation (19)). Thus, with alignment, we have

$$\mathbb{E}[\mathbf{E}_{ti}\mathbf{E}_{si}] = \begin{cases} \epsilon^2, & \text{if t and s are within the same period} \\ 0, & \text{otherwise.} \end{cases} \tag{28}$$

Next, we want to calculate $\mathbb{E}[\mathbf{U}_{tj}\mathbf{U}_{sj}]$, where $\mathbf{U}$ are the left singular vectors of $\mathbf{X}$:

$$\mathbf{X} = \mathbf{U}\mathbf{\Sigma}\mathbf{V}^\top. \tag{29}$$

Recall that after alignment, $\mathbf{X}$ has rank $P$ but is close to a rank-1 matrix. Mathematically, we can rewrite $\mathbf{X} = \mathbf{1}\mathbf{x}_0^\top + \mathbf{C}$, where

$$\mathbf{x}_0 = \begin{bmatrix} x_{1-P} \\ x_{2-P} \\ \ldots \\ x_0 \end{bmatrix} \tag{30}$$

and

$$\mathbf{C} = \begin{bmatrix} e_{1-P} & e_{2-P} & \ldots & e_0 \\ e_{1-P} + e_1 & e_{2-P} & \ldots & e_0 \\ e_{1-P} + e_1 & e_{2-P} + e_2 & \ldots & e_0 \\ \vdots & \vdots & \ddots & \vdots \\ \sum_{i=1} e_{1+(i-2)P} \cdots & \sum_{i=1} e_{2+(i-2)P} & \ldots & \sum_{i=1} e_{(i-1)P} \cdot \end{bmatrix} \tag{31}$$

$\mathbf{x}_0$ is the initial values that are constants. $\mathbf{C}$ is the error matrix where each column is a Gaussian random walk with $P$ duplicates. Thus,

$$\mathbb{E}[\mathbf{X}^\top\mathbf{X}] = \mathbb{E}[(\mathbf{1}\mathbf{x}_0^\top)^\top(\mathbf{1}\mathbf{x}_0^\top) + \mathbf{C}^\top\mathbf{C}] \qquad (\text{because } \mathbb{E}[\mathbf{C}\mathbf{x}_0] = 0) \tag{32}$$

$$= T\mathbf{x}_0\mathbf{x}_0^\top + \mathbb{E}[\mathbf{C}^\top\mathbf{C}] \tag{33}$$

$$= T\mathbf{x}_0\mathbf{x}_0^\top + \frac{T(T+P)\epsilon^2}{2P}\mathbf{I}, \tag{34}$$

where the last equality stands because

$$\mathbb{E}[\mathbf{C}^\top\mathbf{C}]_{ij} = \mathbb{E}[\sum_{k=1} \mathbf{C}_{ki}\mathbf{C}_{kj}] = \begin{cases} 0, \text{ if } j \neq k \text{ because } \mathbf{C}_{ki} \text{ and } \mathbf{C}_{kj} \text{are independent} \\ \sum_{i=1}^T \mathbb{E}[\mathbf{C}_{ij}^2] \approx P\sum_{i=1}^{T/P} i\epsilon^2 = \frac{T(T+P)\epsilon^2}{2P}. \end{cases} \tag{35}$$

The first eigenvectors of $\mathbb{E}[\mathbf{X}^\top\mathbf{X}]$ is $\mathbf{x}_0/||\mathbf{x}_0||_2$ because

$$\mathbb{E}[\mathbf{X}^\top\mathbf{X}]\frac{\mathbf{x}_0}{||\mathbf{x}_0||_2} = (T||\mathbf{x}_0||_2^2 + \frac{T(T+P)\epsilon^2}{2P})\frac{\mathbf{x}_0}{||\mathbf{x}_0||_2}. \tag{36}$$

The corresponding eigenvalue is $(T||\mathbf{x}_0||_2^2 + \frac{T(T+P)\epsilon^2}{2P})$. The rest of the eigenvectors are vectors $\mathbf{w}$ where $\mathbf{w}$ is orthogonal to $\mathbf{x}_0$ because

$$\mathbb{E}[\mathbf{X}^\top\mathbf{X}]\mathbf{w} = \frac{T(T+P)\epsilon^2}{2P}\mathbf{w} \tag{37}$$

with eigenvalues $\frac{T(T+P)\epsilon^2}{2P}$. In summary, we know the right singular vectors $\mathbf{V}$ are

$$\mathbf{V} = \begin{bmatrix} | & | & \cdots & | \\ \frac{\mathbf{x}_0}{||\mathbf{x}_0||_2} & \mathbf{w}_2 & \cdots & \mathbf{w}_P \\ | & | & \cdots & | \end{bmatrix}, \tag{38}$$

where $\mathbf{w}_i$ are orthogonal to $\mathbf{x}_0$.

Next, based on the results of $\mathbf{V}$, we can understand $\mathbf{U}$ by using the formula $\mathbf{X}\mathbf{v}_i = s_i\mathbf{u}_i$, where $s_i$ is the $i$-th singular value and $\mathbf{v}_i, \mathbf{u}_i$ are the $i$-th columns in $\mathbf{V}, \mathbf{U}$, respectively. For the first column ($i = 1$), we know $\mathbf{v}_1 = \frac{\mathbf{x}_0}{||\mathbf{x}_0||_2}$ from Equation (36). Thus,

$$\mathbf{u}_1 = \frac{1}{s_1}\mathbf{X}\frac{\mathbf{x}_0}{||\mathbf{x}_0||_2} \tag{39}$$

$$= \frac{1}{s_1||\mathbf{x}_0||_2}(\mathbf{1}\mathbf{x}_0^\top + \mathbf{C})\mathbf{x}_0 \tag{40}$$

$$= \frac{||\mathbf{x}_0||_2}{s_1}\mathbf{1} + \frac{1}{s_1||\mathbf{x}_0||_2}\mathbf{C}\mathbf{x}_0. \tag{41}$$

For the remaining columns in $\mathbf{U}$, we have

$$\mathbf{u}_i = \frac{1}{s_i}\mathbf{X}\mathbf{w}_i \tag{42}$$

$$= \frac{1}{s_i}(\mathbf{1}\mathbf{x}_0^\top + \mathbf{C})\mathbf{w}_i \tag{43}$$

$$= \mathbf{C}\mathbf{w}_i. \tag{44}$$

In both of the results, we can see that $\mathbf{u}_i$ is a vector of constant ($\frac{||\mathbf{x}_0||_2}{s_i}\mathbf{1}$ if $i = 1$ and $\mathbf{0}$ if $i \geq 2$) plus a linear combination of Gaussian random walk (i.e. $\mathbf{C}$). We know that in a Gaussian random walk, such as the $j$-th column in $\mathbf{C}$, $\mathbb{E}[\mathbf{C}_{tj}\mathbf{C}_{sj}] > 0$. Thus, $\mathbb{E}[\mathbf{U}_{tj}\mathbf{U}_{sj}] > 0$ as well. Combining this with Equation (28), we can see that the second part in Equation (24) is positive. Thus, $\text{RSS}_{\text{w/}} < \text{RSS}_{\text{w/o}}$. To get a lower-bound estimate, we can just count the effect of $\mathbf{u}_1$ and assume that $\mathbf{u}_1 = \frac{1}{\sqrt{T}}\mathbf{1}$, then the second part in Equation (24) is roughly

$$\underbrace{P}_{\text{iteration of }i} \cdot \overbrace{1}^{\text{just count } u_1} \cdot \underbrace{T(P-1)\epsilon^2}_{\text{Equation 28 and } \sum_{t=1}^{T}\sum_{s=1,s\neq t}^{T}\mathbf{E}_{ti}\mathbf{E}_{si}} \overbrace{\frac{1}{T}}^{\mathbf{U}_{tj}\mathbf{U}_{sj}} = P(P-1)\epsilon^2. \tag{45}$$

Thus, the upper-bound of $\text{RSS}_{\text{w/}}$ is

$$\text{RSS}_{\text{w/}} \lesssim TP\epsilon^2 - P^2\epsilon^2 - P(P-1)\epsilon^2 = (T - 2P + 1)P\epsilon^2 < \text{RSS}_{\text{w/o}}. \tag{46}$$

## A.2 INCREASE IN TESTING ERROR

While the training error is lower after alignment, the testing error is higher. Intuitively, this is because with alignment, the noise in the data is also easier to learn and therefore resulted in "overfitting."

Mathematically, to estimate the testing error, we start with

$$\text{testing error} = \mathbb{E}[\|\mathbf{z}^\top \hat{\mathbf{B}} - (\mathbf{z} + \mathbf{e})^\top\|_F^2] \tag{47}$$

$$= \mathbb{E}[\|\mathbf{z}^\top (\mathbf{I} + (\mathbf{X}^\top \mathbf{X})^{-1}\mathbf{X}^\top \mathbf{E}) - (\mathbf{z} + \mathbf{e})^\top\|_F^2] \quad \text{(from Equation (8))} \tag{48}$$

$$= \mathbb{E}[\|\mathbf{z}^\top (\mathbf{X}^\top \mathbf{X})^{-1}\mathbf{X}^\top \mathbf{E} - \mathbf{e}^\top\|_F^2] \tag{49}$$

$$= \mathbb{E}\left[\text{tr}\left((\mathbf{E}^\top \mathbf{X}(\mathbf{X}^\top \mathbf{X})^{-1}\mathbf{z} - \mathbf{e})(\mathbf{z}^\top (\mathbf{X}^\top \mathbf{X})^{-1}\mathbf{X}^\top \mathbf{E} - \mathbf{e}^\top)\right)\right] \tag{50}$$

$$= \mathbb{E}\left[\|\mathbf{e}\|_F^2\right] + \text{tr}\left(\mathbf{E}^\top \mathbf{X}(\mathbf{X}^\top \mathbf{X})^{-1}\mathbb{E}[\mathbf{z}\mathbf{z}^\top](\mathbf{X}^\top \mathbf{X})^{-1}\mathbf{X}^\top \mathbf{E}\right) \tag{51}$$

$$= \mathbb{E}\left[\|\mathbf{e}\|_F^2\right] + \text{tr}\left(\mathbf{E}^\top \mathbf{X}(\mathbf{X}^\top \mathbf{X})^{-1}\frac{1}{T}\mathbf{X}^\top \mathbf{X}(\mathbf{X}^\top \mathbf{X})^{-1}\mathbf{X}^\top \mathbf{E}\right) \tag{52}$$

$$= \mathbb{E}\left[\|\mathbf{e}\|_F^2\right] + \frac{1}{T}\text{tr}(\mathbf{E}\mathbf{E}^\top \mathbf{P}) \tag{53}$$

$$= \mathbb{E}\left[\|\mathbf{e}\|_F^2\right] + \frac{1}{T}\text{tr}(\mathbf{E}\mathbf{E}^\top \mathbf{U}\mathbf{U}^\top) \tag{54}$$

$$= \mathbb{E}\left[\|\mathbf{e}\|_F^2\right] + \frac{1}{T}\|\mathbf{E}^\top \mathbf{U}\|_F^2, \tag{55}$$

where $\mathbf{z}$ is a sample vector from the testing distribution (which is assumed to the same as the training distribution) and $\mathbf{e}$ is the corresponding noise vector. Notice that this equation is similar to Equation (18) where the first part of the expression is the same with or without alignment. However, here the two parts are summed together, whereas in Equation (18), the second part is subtracted from the first part. Since the second part is essentially the same (except a factor of $\frac{1}{T}$), we can see that the testing error with alignment is higher by the following the same derivation in the previous section. To be precise, the testing errors are

$$\text{RSS}_{\text{w/o}} \approx P(1 + \frac{P}{T})\epsilon^2 \text{ and } \text{RSS}_{\text{w/}} \gtrsim P(1 + \frac{2P+1}{T})\epsilon^2. \tag{56}$$

### A.3 LESS INCREASE IN TRAINING ERROR WITH REDUCED-RANK REGRESSION (RRR)

Under this setting, we can see that the fitted solution $\hat{\mathbf{B}} = \mathbf{I} + (\mathbf{X}^\top \mathbf{X})^{-1}\mathbf{X}^\top \mathbf{E} \in \mathbb{R}^P$ (in Equation (8)) is of full rank. However, a quadratic matrix is quite parameter heavy, so sometimes we want to use a lower-rank approximation to reduce model size and overfitting. This is called reduced-rank regression (RRR), where the parameter matrix $\mathbf{B}$ is also subject to a rank constraint:

$$\hat{\mathbf{B}}_{\text{RRR}} = \min_{\mathbf{B}}\|\mathbf{Y} - \mathbf{X}\mathbf{B}\|_F^2, \text{ where } \text{rank}(\mathbf{B}) \leq r. \tag{57}$$

In this section, we show that alignment helps mitigate the increase of training loss in RRR.

First, we know that Equation (57) is equivalent to

$$\hat{\mathbf{B}}_{\text{RRR}} = \min_{\mathbf{B}}\|\mathbf{Y} - \mathbf{X}\hat{\mathbf{B}}\|_F^2 + \|\mathbf{X}\hat{\mathbf{B}} - \mathbf{X}\mathbf{B}\|_F^2, \tag{58}$$

because linear regression is essentially an orthogonal projection of $\mathbf{Y}$ onto the column space of $\mathbf{X}$. The first term in the minimization does not depend on $\mathbf{B}$, so we only need to minimize the second term. Based on the Eckart–Young–Mirsky theorem (E. Schmidt, 1907), the solution is

$$\hat{\mathbf{B}}_{\text{RRR}} = \hat{\mathbf{B}}\mathbf{V}_r\mathbf{V}_r^\top, \tag{59}$$

where $\mathbf{V}_r$ are the first $r$ right-singular vectors of $\mathbf{X}\hat{\mathbf{B}}$. In other words, $\mathbf{X}\hat{\mathbf{B}}_{\text{RRR}}$ is the best rank-$r$ approximation of $\mathbf{X}\hat{\mathbf{B}}$ under Frobenius norm. Thus, based on the Eckart–Young–Mirsky theorem (E. Schmidt, 1907),

$$\|\mathbf{X}\hat{\mathbf{B}} - \mathbf{X}\mathbf{B}\|_F^2 = \sum_{i=r+1}^{P} s_i^2, \tag{60}$$

where $s_i$ is the $i$-th singular value of $\mathbf{X}\hat{\mathbf{B}}$. To find $s_i$, we calculate the eigenvalues of $\mathbf{X}\hat{\mathbf{B}}$ following the derivation below:

$$\mathbb{E}[(\mathbf{X}\hat{\mathbf{B}})^\top \mathbf{X}\hat{\mathbf{B}}] = \mathbb{E}[(\mathbf{I} + (\mathbf{X}^\top \mathbf{X})^{-1}\mathbf{X}^\top \mathbf{E})^\top \mathbf{X}^\top \mathbf{X}(\mathbf{I} + (\mathbf{X}^\top \mathbf{X})^{-1}\mathbf{X}^\top \mathbf{E})] \tag{61}$$

$$= \mathbb{E}[\mathbf{X}^\top \mathbf{X} + \mathbf{E}^\top \mathbf{X} + \mathbf{X}^\top \mathbf{E} + \mathbf{E}^\top \mathbf{X}(\mathbf{X}^\top \mathbf{X})^{-1}\mathbf{X}^\top \mathbf{E}] \tag{62}$$

$$= \mathbb{E}[\mathbf{X}^\top \mathbf{X} + \mathbf{E}^\top \mathbf{X}(\mathbf{X}^\top \mathbf{X})^{-1}\mathbf{X}^\top \mathbf{E}] \tag{63}$$

$$\approx \mathbf{X}^\top \mathbf{X}, \tag{64}$$

where the last approximation stands because the noise terms $\mathbf{E}$ should have much smaller magnitude than the values $\mathbf{X}$. Thus, combining Equation (60) and (64), we want to compare the eigenvalues of $\mathbf{X}^\top \mathbf{X}$ with or without alignment in order to understand which one is better under RRR.

Intuitively, with alignment, $\mathbf{X}^\top \mathbf{X}$ is close to a rank-1 matrix, so there is only one non-zero eigenvalue. In contrast, without alignment, the eigenvalues are all non-zeros. Note that the sum of all squared singular values of $\mathbf{X}$ are the same with or without alignment because they have the same Frobenius norm. This implies that the single non-zero eigenvalue under alignment equals to the sum of all eigenvalues under no alignment and thus according to Equation (60), with alignment results in better RRR performance. However, this is just a rough estimate. Then next two sections will give a much precise estimation of the eigenvalues.

### A.3.1 RRR WITHOUT ALIGNMENT

Similar to Equation (30) and(31), we can also split $\mathbf{X}$ into two parts $\mathbf{X} = \mathbf{H} + \mathbf{C}$, where

$$\mathbf{H} = \begin{bmatrix} x_{1-P} & x_{2-P} & \dots & x_{-1} & x_0 \\ x_{2-P} & x_{3-P} & \dots & x_0 & x_{1-P} \\ x_{3-P} & x_{4-P} & \dots & x_{1-P} & x_{2-P} \\ \vdots & \vdots & \ddots & \vdots & \vdots \\ x_0 & x_{1-P} & \dots & x_{-2} & x_{-1} \\ x_{1-P} & x_{2-P} & \dots & x_{-1} & x_0 \\ \vdots & \vdots & \vdots & \vdots & \vdots \end{bmatrix} \tag{65}$$

and

$$\mathbf{C} = \begin{bmatrix} e_{1-P} & e_{2-P} & \dots & e_{-1} & e_0 \\ e_{2-P} & e_{3-P} & \dots & e_0 & e_{1-P} + e_1 \\ e_{3-P} & e_{4-P} & \dots & e_{1-P} + e_1 & e_{2-P} + e_2 \\ \vdots & \vdots & \vdots & \vdots & \vdots \end{bmatrix}. \tag{66}$$

Notice that $\mathbf{H}$ is a non-square circulant Hankel (or anti-circulant) matrix starting from $\mathbf{x}_0$ where all elements on any anti-diagonals are the same.

To roughly estimate the singular values of $\mathbf{X}$, we can first assume that $\mathbb{E}[x_i, x_j] = 0, \forall i \neq j$, where $x_i, x_j \in \mathbf{x}_0$. Then,

$$\mathbb{E}[\mathbf{X}^\top \mathbf{X}] = \mathbb{E}[\mathbf{H}^\top \mathbf{H} + \mathbf{C}^\top \mathbf{C}] \tag{67}$$

$$= \frac{T\|\mathbf{x}_0\|_2^2}{P}\mathbf{I} + \frac{T(T+P)\epsilon^2}{2P}\mathbf{I} \tag{68}$$

$$= (\frac{T\|\mathbf{x}_0\|_2^2}{P} + \frac{T(T+P)\epsilon^2}{2P})\mathbf{I}, \tag{69}$$

where $\mathbb{E}[\mathbf{C}^\top \mathbf{C}]$ is calculated using Equation (35). Thus, we can see that the eigenvalues are all $\frac{T\|\mathbf{x}_0\|_2^2}{P} + \frac{T(T+P)\epsilon^2}{2P}$.

However, this is an very rough estimate since the assumption of $\mathbb{E}[x_i, x_j] = 0$ is usually not the case. To be exact, first let

$$\mathbf{H}' = \begin{bmatrix} x_{1-P} & x_{2-P} & \dots & x_{-1} & x_0 \\ x_{2-P} & x_{3-P} & \dots & x_0 & x_{1-P} \\ x_{3-P} & x_{4-P} & \dots & x_{1-P} & x_{2-P} \\ \vdots & \vdots & \ddots & \vdots & \vdots \\ x_{-1} & x_0 & \dots & x_{-3} & x_{-2} \\ x_0 & x_{1-P} & \dots & x_{-2} & x_{-1} \end{bmatrix} \in \mathbb{R}^{P \times P}, \tag{70}$$

which is a square circulant Hankel matrix. Then,

$$\mathbf{H} = \left.\begin{bmatrix} \mathbf{H'} \\ \vdots \\ \mathbf{H'} \end{bmatrix}\right\} \text{ repeat } \frac{T}{P} \text{ times.} \tag{71}$$

Thus, $\mathbf{H}^\top \mathbf{H} = \frac{T}{P}(\mathbf{H'})^2$. The eigenvalues $\lambda'_k$ of $\mathbf{H'}$ can be calculated using Lemma (A.1). Then the eigenvalues $\lambda_i$ of $\mathbf{H}^\top \mathbf{H}$ is $\lambda_i = \frac{T}{P}(\lambda'_i)^2$. We can see that the eigenvalues are thus non-zeros. For example,

$$\lambda_0 = \frac{T}{P} \left( \sum_{j=0}^{P-1} x_{j-P+1} \right)^2. \tag{72}$$

Combine this with the rough estimation in Equation (69) and assuming that $x_j$ are random, we can say that the eigenvalues are scattered around $\frac{T\|\mathbf{x}_0\|_2^2}{P} + \frac{T(T+P)\epsilon^2}{2P}$, which can be used to estimate the RRR residual following Equation (60).

**Lemma A.1.** *If matrix $\mathbf{A} \in \mathbb{R}^{n \times n}$ is a square circulant Hankel matrix with full rank and the first row is $\mathbf{a}^\top = [a_0, \ldots, a_{n-1}]^\top$, then its eigenvalues are*

$$\lambda_k = \begin{cases} \mathbf{F}_{\mathbf{a},0}, & \text{if } k = 0, \\ \mathbf{F}_{\mathbf{a},n/2} \text{ or } -\mathbf{F}_{\mathbf{a},n/2}, & \text{if } n \text{ is even and } k = n/2 \\ \pm\sqrt{\mathbf{F}_{\mathbf{a},k}\mathbf{F}_{\mathbf{a},n-k}}, & \text{otherwise.} \end{cases} \tag{73}$$

*where*

$$\mathbf{F}_{\mathbf{a},k} = \sum_{j=0}^{n-1} a_j \exp(\frac{-kj}{n} 2\pi i). \tag{74}$$

*Proof.* We have a circulant Hankel matrix

$$\mathbf{A} = \begin{bmatrix} a_{00} & a_{01} & \cdots & a_{0(n-1)} \\ a_{10} & a_{11} & \cdots & a_{1(n-1)} \\ \vdots & \vdots & \ddots & \vdots \\ a_{(n-1)0} & a_{(n-1)1} & \cdots & a_{(n-1)(n-1)} \end{bmatrix} = \begin{bmatrix} a_0 & a_1 & \cdots & a_{n-1} \\ a_1 & a_2 & \cdots & a_0 \\ \vdots & \vdots & \ddots & \vdots \\ a_{n-1} & a_0 & \cdots & a_{n-2} \end{bmatrix}, \tag{75}$$

where any anti-diagonals have the same values and $a_{ij} = a_{(i+j)(\text{mod } n)}$.

Next, we define the following discrete Fourier Transform (DFT) on a vector $\mathbf{v} = [v_0, \ldots, v_{n-1}]^\top$:

$$\mathcal{F}(\mathbf{v})_k = \sum_{j=0}^{n-1} v_j \exp(\frac{-kj}{n} 2\pi i) = \mathbf{Fv}, \tag{76}$$

where $\mathbf{F}$ is the unnormalized DFT matrix with properties $\mathbf{F}^\top = \mathbf{F}$ and $\mathbf{F}^{-1} = \frac{1}{n}\mathbf{F}^*$.

Then,

$$\mathcal{F}(\mathbf{Av})_k = \sum_{j=0}^{n-1} (\mathbf{Av})_j \exp(\frac{-kj}{n} 2\pi i) \tag{77}$$

$$= \sum_{j=0}^{n-1} \left( \sum_{l=0}^{n-1} a_{(l+j)(\mathrm{mod}\, n)} v_l \right) \exp(\frac{-k(j+l)}{n} 2\pi i) \exp(\frac{kl}{n} 2\pi i) \tag{78}$$

$$= \sum_{l=0}^{n-1} v_l \exp(\frac{kl}{n} 2\pi i) \left( \sum_{j=0}^{n-1} a_{(l+j)(\mathrm{mod}\, n)} \exp(\frac{-k(j+l)}{n} 2\pi i) \right) \tag{79}$$

$$= \sum_{l=0}^{n-1} v_l \exp(\frac{kl}{n} 2\pi i) \left( \sum_{j=-l}^{n-l-1} a_{l+j} \exp(\frac{-k(j+l)}{n} 2\pi i) \right) \tag{80}$$

$$= \sum_{l=0}^{n-1} v_l \exp(\frac{kl}{n} 2\pi i) \sum_{j'=0}^{n-1} a_{j'} \exp(\frac{-kj'}{n} 2\pi i) \tag{81}$$

$$= \mathcal{F}^*(\mathbf{v})_k \mathcal{F}(\mathbf{a})_k, \tag{82}$$

where $\mathbf{a} = [a_0, \ldots, a_{n-1}]^\top$. For $\mathbf{v}$ to be an eigenvector of $\mathbf{A}$, we need $\mathbf{Av} = \lambda\mathbf{v}$, which implies

$$\mathcal{F}(\mathbf{a})_k \mathcal{F}^*(\mathbf{v})_k = \lambda \mathcal{F}(\mathbf{v})_k. \tag{83}$$

Collecting all $k$, the equation is equivalently

$$\mathbf{F_a F^* v} = \lambda \mathbf{Fv}, \tag{84}$$

where $\mathbf{F_a} = \mathrm{diag}(\mathbf{Fa})$. Thus, the eigenvalue $\lambda$ satisfies

$$\det(\mathbf{F_a F^*} - \lambda \mathbf{F}) \Leftrightarrow \det(\mathbf{F_a} - \lambda \frac{1}{n} \mathbf{FF}^\top), \tag{85}$$

where

$$\frac{1}{n} \mathbf{FF}^\top = \begin{bmatrix} 1 & 0 & 0 & \ldots & 0 & 0 \\ 0 & 0 & 0 & \ldots & 0 & 1 \\ 0 & 0 & 0 & \ldots & 1 & 0 \\ \vdots & \vdots & \vdots & \ddots & \vdots & \vdots \\ 0 & 0 & 1 & \ldots & 0 & 0 \\ 0 & 1 & 0 & \ldots & 0 & 0 \end{bmatrix}. \tag{86}$$

By decomposing into block matrices, from Equation (85), eigenvalues satisfy

$$|(\lambda - \mathbf{F}_{\mathbf{a},0})| \det(\mathbf{F'_a} - \lambda \mathbf{J}) = 0 \Leftrightarrow |(\lambda - \mathbf{F}_{\mathbf{a},0})| \det(\mathbf{JF'_a} - \lambda \mathbf{I}) = 0, \tag{87}$$

where

$$\mathbf{J} = \begin{bmatrix} 0 & 0 & \ldots & 0 & 1 \\ 0 & 0 & \ldots & 1 & 0 \\ \vdots & \vdots & \ddots & \vdots & \vdots \\ 0 & 1 & \ldots & 0 & 0 \\ 1 & 0 & \ldots & 0 & 0 \end{bmatrix} \in \mathbb{R}^{(n-1)\times(n-1)}. \tag{88}$$

is the exchange matrix and $\mathbf{F'_a} = \mathrm{diag}([\mathbf{F}_{\mathbf{a},1}, \ldots, \mathbf{F}_{\mathbf{a},n-1}])$. In addition to $\lambda_0 = \mathbf{F}_{\mathbf{a},0}$, we also need to find the eigenvalues of $\mathbf{JF'_a}$. We can see that

$$\mathbf{JF'_a} = \begin{bmatrix} 0 & 0 & \ldots & 0 & \mathbf{F}_{\mathbf{a},n-1} \\ 0 & 0 & \ldots & \mathbf{F}_{\mathbf{a},n-2} & 0 \\ \vdots & \vdots & \ddots & \vdots & \vdots \\ 0 & \mathbf{F}_{\mathbf{a},2} & \ldots & 0 & 0 \\ \mathbf{F}_{\mathbf{a},1} & 0 & \ldots & 0 & 0 \end{bmatrix} \tag{89}$$

Table 2: Mean-Squared Errors (MSEs) on the test set of two variations of PAPER for three datasets and four horizons. Mean and standard deviation reported over 10 runs. Shaded number indicates the best performing model and is superscribed with † if the outperformance is statistically significant with $p$-value less than $5\%$.

| Dataset | Horizon | SFNN + rolling PAPER | SFNN + PAPER |
|---|---|---|---|
| Electricity | 168 | $0.1620 \pm 0.0005$ | $0.1589^{\dagger} \pm 0.0002$ |
| | 336 | $0.1640 \pm 0.0002$ | $0.1591^{\dagger} \pm 0.0002$ |
| | 504 | $0.1707 \pm 0.0003$ | $0.1674^{\dagger} \pm 0.0004$ |
| | 672 | $0.1843 \pm 0.0010$ | $0.1836 \pm 0.0003$ |
| Solar | 144 | $0.2193 \pm 0.0070$ | $0.2047^{\dagger} \pm 0.0052$ |
| | 288 | $0.2092 \pm 0.0027$ | $0.2020^{\dagger} \pm 0.0028$ |
| | 432 | $0.2163 \pm 0.0024$ | $0.2090^{\dagger} \pm 0.0015$ |
| | 576 | $0.2296 \pm 0.0032$ | $0.2239 \pm 0.0015$ |
| Traffic | 168 | $0.3387 \pm 0.0016$ | $0.3319^{\dagger} \pm 0.0015$ |
| | 336 | $0.3409 \pm 0.0009$ | $0.3384^{\dagger} \pm 0.0009$ |
| | 504 | $0.3455 \pm 0.0009$ | $0.3430 \pm 0.0001$ |
| | 672 | $0.3570 \pm 0.0005$ | $0.3589 \pm 0.0007$ |

is an anti-diagonal matrix, so we have

$$(\mathbf{JF'_a})^2 = \begin{bmatrix} \mathbf{F_{a,1}F_{a,n-1}} & 0 & \dots & 0 & 0 \\ 0 & \mathbf{F_{a,2}F_{a,n-2}} & \dots & 0 & 0 \\ \vdots & \vdots & \ddots & \vdots & \vdots \\ 0 & 0 & \dots & \mathbf{F_{a,n-2}F_{a,2}} & 0 \\ 0 & 0 & \dots & 0 & \mathbf{F_{a,n-1}F_{a,1}} \end{bmatrix}. \tag{90}$$

Combining this with Equation (87), we have the eigenvalues as

$$\lambda_k = \begin{cases} \mathbf{F_{a,0}}, & \text{if } k = 0, \\ \mathbf{F_{a,n/2}} \text{ or } -\mathbf{F_{a,n/2}}, & \text{if } n \text{ is even and } k = n/2 \\ \pm\sqrt{\mathbf{F_{a,k}F_{a,n-k}}}, & \text{otherwise.} \end{cases} \tag{91}$$

$\square$

### A.3.2 RRR WITH ALIGNMENT

We have already calculated the eigenvalues of $\mathbf{X}^{\top}\mathbf{X}$ in Section A.1.2. In which, we showed that

$$\lambda_k = \begin{cases} (T\|\mathbf{x}_0\|_2^2 + \frac{T(T+P)\epsilon^2}{2P}, & \text{if } k = 0, \\ \frac{T(T+P)\epsilon^2}{2P}, & \text{otherwise.} \end{cases} \tag{92}$$

Note that the sum of all eigenvalues are the same with or without alignment. However, with alignment, $T\|\mathbf{x}_0\|_2^2$ concentrates its contribution only to $\lambda_0$, whereas without alignment, its contribution are spread out across all eigenvalues.

## B RESULTS FOR ABLATION STUDY

See Table 2.

## C RESULTS FOR MULTIPLE PERIODICTIES

See Table 3.

## D USE OF LARGE LANGUAGE MODELS

This paper was edited for grammar, style, and readability with the assistance of a large language models.

Table 3: Mean-Squared Errors (MSEs) on the test set with and without PAPER for four horizons on two datasets with multiple fundamental periodicties. Mean and standard deviation reported over 10 runs. Shaded number indicates the best performing model and is superscribed with † if the outperformance is statistically significant with $p$-value less than $5\%$.

| Dataset | Horizon | SFNN | SFNN + PAPER |
|---------|---------|------|--------------|
| ETTm1 | 96 | $0.3082^{\dagger} \pm 0.0004$ | $0.3280 \pm 0.0006$ |
| | 192 | $0.3412^{\dagger} \pm 0.0003$ | $0.3455 \pm 0.0005$ |
| | 288 | $0.3641 \pm 0.0003$ | $0.3638 \pm 0.0002$ |
| | 384 | $0.3738 \pm 0.0002$ | $0.3720^{\dagger} \pm 0.0005$ |
| ETTh1 | 168 | $0.3186^{\dagger} \pm 0.0012$ | $0.3226 \pm 0.0009$ |
| | 336 | $0.3510^{\dagger} \pm 0.0025$ | $0.3562 \pm 0.0025$ |
| | 504 | $0.4047 \pm 0.0026$ | $0.4072 \pm 0.0017$ |
| | 672 | $0.4578 \pm 0.0036$ | $0.4444^{\dagger} \pm 0.0023$ |

