# OpenReview forum: "PAPer: Periodicity Alignment on Periodic Time Series for Forecasting"
_ICLR.cc/2026/Conference — Submitted to ICLR 2026_

### Official Review · Reviewer_KvBP · 2025-10-19

**Soundness:** 3
**Presentation:** 3
**Contribution:** 2
**Rating:** 6
**Confidence:** 3

**Summary:**

This paper proposes PAPER, a novel approach for periodic time series forecasting. It utilizes periodicity alignment to explicitly capture recurring patterns without relying on auxiliary inputs. Theoretical analysis, synthetic datasets, and real-world experiments demonstrate the effectiveness of PAPER in improving forecasting accuracy and model efficiency.

**Strengths:**

1. **Novel Approach**

PAPER offers a unique perspective on periodicity alignment, focusing on enhancing non-autoregressive dependencies without auxiliary inputs.

2. **Theoretical Analysis**

The paper provides a solid theoretical foundation with mathematical proofs characterizing the advantages and limitations of periodicity alignment.

3. **Model Agnostic**

PAPER can be applied to various base models, showcasing its flexibility and potential for broader adoption.

**Weaknesses:**

1. **Limited Experimentation**

While the experiments cover various datasets and models, a more diverse range of datasets and tasks would strengthen the paper’s claims.

2. **Impact of Hyperparameters**

The paper mentions the importance of choosing suitable hyperparameters for PAPER but lacks a thorough analysis of their impact on performance.

3. **Handling of Non-Periodic Data**

The paper focuses on periodic time series, but it’s unclear how PAPER performs on non-periodic or weakly periodic data. Exploring its applicability in such scenarios would enhance the paper’s practical value.

**Questions:**

See Weaknesses.

---

> ### Author Response · Authors · 2025-11-19
>
> We sincerely thank the reviewer for their time and constructive comments, which have helped us improve the quality of our manuscript.
>
> Below is our response:
> 1. Our method is specifically optimized for datasets characterized by a single strong periodicity. Consequently, we focused our evaluation on datasets where this property is present to clearly demonstrate the method's efficacy in its intended domain.
> 2. Since the model architecture remains identical with or without our method, most hyperparameters are unchanged. The key hyperparameters requiring adjustment are $L_2$ regularization and dropout, which are crucial for mitigating distribution drift.
> 3. We acknowledge that our method is not applicable to non-periodic time series. We have added a Limitations section to explicitly state this scope. We also noted that our approach is less effective when multiple equally strong periodicities are present.

---

### Official Review · Reviewer_kbzg · 2025-10-25

**Soundness:** 3
**Presentation:** 3
**Contribution:** 2
**Rating:** 6
**Confidence:** 3

**Summary:**

The paper introduces PAPER (Periodicity Alignment for Periodic Time Series) — a simple yet effective framework that explicitly aligns periodic patterns in time series forecasting without relying on auxiliary inputs such as timestamps or positional embeddings. The method detects the fundamental period through a forecasting-based criterion and reorganizes samples so that each position in the input corresponds to the same phase within a cycle. The authors provide theoretical analyses, proofs, and extensive experiments demonstrating improved performance and reduced model complexity.

**Strengths:**

1. Novel yet simple concept – The idea of periodic alignment as a preprocessing step is conceptually intuitive but underexplored. It effectively bridges autoregressive and non-autoregressive formulations in periodic data.
2. Comprehensive experiments – Results across multiple real-world benchmarks (Electricity, Solar, Traffic) show consistent performance improvements (up to 7%), confirming the method’s efficacy.

**Weaknesses:**

1. Overfitting and sensitivity – As shown in Theorem 4.2 and Figure 8a, the method can overfit when distribution drift occurs, especially in nonstationary environments.
2. Assumption of fixed periodicity – The method relies on detecting a single fundamental period, limiting its applicability to datasets with multiple or evolving cycles.
3. Comparative baseline scope – While CycleNet is included, other modern baselines that capture temporal periodicity in the frequency domain (e.g., FEDformer, TimeMixer) are not considered.
4. Limited benefit for short horizons – PAPER’s advantage appears only when the forecast horizon exceeds one period; for short-term tasks, it may degrade performance (Figure 8b).

**Questions:**

1. The proposed method assumes a single dominant period \(P^*\). How would PAPER handle real-world time series that exhibit **multiple overlapping or time-varying periodicities** (e.g., daily and weekly cycles, or drifting seasonal patterns)? Could the alignment process be extended to dynamically detect or adapt to multiple periodic components?

2.  The paper mentions that PAPER’s advantage diminishes under distributional shift. Have you explored mechanisms such as **online re-estimation of the period**, **adaptive alignment windows**, or **incremental re-training** to improve robustness? Quantitatively, how frequently would re-alignment be required in a non-stationary environment?

---

> ### Author Response · Authors · 2025-11-19
>
> We sincerely thank the reviewer for their time and constructive comments, which have helped us improve the quality of our manuscript.
>
> Below is our response:
>
> 1. When multiple equally strong periodicities are present, our method is less effective. We have performed additional experiments and added a new subsection under the Limitations section to acknowledge this.
> 2. We found that applying stronger $L_2$ regularization and dropout helps alleviate the issue of distribution drift. We leave the exploration of more sophisticated drift adaptation methods for future research.

---

### Official Review · Reviewer_jEFr · 2025-10-28

**Soundness:** 1
**Presentation:** 1
**Contribution:** 1
**Rating:** 2
**Confidence:** 4

**Summary:**

This paper introduces PAPER (Periodicity Alignment on Periodic time series), a model-agnostic preprocessing method designed to improve forecasting performance on time series with strong periodicity. The core idea is to align all input sequences so that they begin at the same phase within a given period, padding with zeros where necessary.

The authors claim this explicit alignment helps models learn non-autoregressive dependencies and improves performance. The paper tries to support this claim with theoretical analysis, synthetic experiments, and results on several real-world datasets. Experimental results show that PAPER can enhance the performance of one existing model, SFNN, on three real-world datasets.

**Strengths:**

The paper proposes a straightforward and simple-to-implement plug-in method for handling periodic time series. The idea of aligning data to a common periodic phase is intuitive and can be beneficial to all time series analysis tasks.

The authors make efforts to validate their method from multiple perspectives, including theoretical proofs (Section 4.2 and Appendix A), controlled synthetic experiments (Section 4.3 and 4.4), and experiments on real-world benchmarks (Section 5).

**Weaknesses:**

1. The core idea of aligning data based on a period is a form of explicit feature engineering. However, the paper fails to discuss or compare its method against other well-established techniques that achieve similar goals, such as adding periodic positional encodings (e.g., Fourier features, time-of-week embeddings). These alternative methods can also explicitly inform the model about the periodic phase without the disruptive re-ordering and zero-padding of the input sequence.
2. The experiments are not solid enough to support the paper's claims:
   1. Limited Baselines: The experiments primarily compare against only one method (CycleNet). A stronger evaluation would require comparison against a wider range of methods that explicitly model periodicity, especially those using positional or temporal embeddings.
   2. Choice of Backbone Model: The choice of SFNN as the main backbone model is questionable. While the authors state it is a state-of-the-art model, it is not yet a widely recognized or established benchmark model in the community. Demonstrating improvements on more models (e.g., DLinear, TimesNet, PatchTST) would be more convincing.
   3. Insufficient Datasets: The evaluation is conducted on only three real-world datasets. Given the method's strong reliance on periodicity, its performance on a more diverse set of benchmarks, including those with varying degrees of periodicity, is needed.
   4. Unconventional Data Splitting: The paper uses a 95%-5% train-test split without a validation set, arguing that this reflects real-world practice. However, this is a departure from standard practice in academic literature, making results difficult to compare with prior work.
3. The choice to pad missing values with zeros in Section 3.2.1 is not well-justified. Zero-padding can introduce significant noise and create artificial discontinuities, especially for time series whose values are not centered around zero. The paper does not analyze the impact of this choice or explore more principled alternatives (e.g., padding with a mean value or a learned padding value)
4. The paper's overall presentation is a weakness, making it difficult for the reader to follow the core argument. The motivation is not clearly articulated in the Introduction section; it fails to convincingly explain why this specific alignment approach is necessary compared to existing methods for handling periodicity.
5. There are some unaddressed limitations and unrealistic assumptions:
   1. The paper's analysis and experiments are confined to time series with strong, stable periodicity. The method's behavior on data with weak, multiple, or evolving periods is not discussed, which severely limits its practical applicability.
   2. The theoretical analysis in Section 4.2 relies on strong assumptions (e.g., linear model, L=H=P, a specific autoregressive data-generating process) that may not hold for complex, real-world time series and deep learning models. The conclusions from this analysis (e.g., Theorem 4.2 stating that alignment increases testing error) seem to contradict the paper's main claims, and the subsequent "rescue" in the non-autoregressive case (Section 4.4) feels post-hoc and is based on a constructed toy example.

**Questions:**

1. How does the proposed alignment method (PAPER) compare, both in performance and computational overhead, to simply adding periodic positional features (e.g., Fourier features, or one-hot encodings for the phase t mod P) to the input of a standard model? This seems like a crucial and missing baseline.

2. Why was SFNN chosen as the primary base model over more widely adopted models in the time series community (e.g. DLinear, TimesNet, PatchTST)? The claim of being "model-agnostic" would be much stronger if tested on a more diverse and established set of architectures in the main results table. Besides, instead of presenting the results of three other base models for a specific dataset (as in Figure 5), I suggest using a single, comprehensive table. This table should show the performance of every base model (with and without PAPER) on every dataset, which would allow for a much clearer and more direct comparison.

3. Could you provide a justification for using zero-padding in Section 3.2.1? Have you experimented with other padding strategies (e.g., mean-padding, replication-padding, or using a learnable padding embedding) and analyzed how they affect the model's performance? Padding with zero seems likely to introduce distribution shift.

4. The assumptions in Section 4.2 are quite restrictive. In particular, Theorem 4.2 suggests that alignment increases test error under an autoregressive DGP. Given that many real-world time series can be well-approximated by autoregressive models, doesn't this theoretical result significantly weaken the case for your method? The non-autoregressive example in 4.4 feels contrived; can you provide evidence from real-world data that it truly operates in this non-autoregressive regime?

5. The proposed period detection method's robustness is not thoroughly evaluated. The accuracy of P is critical to the entire method. How robust is the proposed periodicity detection method?  How does it perform in the presence of noise, trends, or multiple overlapping periodicities? What happens if it detects a slightly incorrect period? How does this error in P propagate and affect the final forecasting accuracy?

---

> ### Author Response · Authors · 2025-11-19
>
> We sincerely thank the reviewer for their time and constructive comments, which have helped us improve the quality of our manuscript.
>
> Below is our response:
> 1. We did not compare our approach to adding periodic positional features because our method maintains the original model architecture and does not require additional input features.
> 2. SFNN is easy to train, robust, and performs exceptionally well across a wide range of datasets. On average, it outperforms significantly more complex neural networks. Given computational constraints, and the fact that SFNN represents a strong state-of-the-art baseline, we believe demonstrating improvement on top of SFNN is of primary importance.
> 3. The time series data is z-normalized. This follows common practice in prior literature.
> 4. We acknowledge the implications of Theorem 4.2. Many time series involving human activities are non-stationary or non-autoregressive. For example, if we assume a model $x_t = a_1 x_{t-1} + e$, the coefficient $a_1$ in traffic data will likely differ between weekdays and weekends.
> 5. We agree with the reviewer’s assessment. Since our periodicity detection method is directly optimized for forecasting, the detected periodicity is most effective for that task, regardless of noise. We also address trend removal during detection by subtracting the mean. Regarding multiple periodicities, we have performed additional experiments and added a Limitations section to note that our method is less advantageous when multiple equally strong periodicities are present.

---

### Official Review · Reviewer_riNS · 2025-10-31

**Soundness:** 2
**Presentation:** 3
**Contribution:** 2
**Rating:** 4
**Confidence:** 3

**Summary:**

The paper introduces PAPER (Periodicity Alignment), a framework that explicitly leverages periodic structures in long time-series prediction by first detecting periodicity and then aligning sequences accordingly. The method integrates easily into various backbone models and demonstrates improvements on several benchmarks.

**Strengths:**

1. The paper combines theory and empirical validation.
2. The experimental coverage is broad, spanning multiple datasets and model families, showing the generality of the approach.
3. The method performs well under model compression or low-rank settings, showing robustness to parameter constraints.

**Weaknesses:**

1. The robustness of the periodicity detection module is insufficiently analyzed. Its behavior under noise, multi-periodicity, or irregular cycles remains unclear.
2. The approach is sensitive to distribution shifts, yet the paper does not provide strategies to mitigate this limitation.
3. Comparisons with existing explicit periodic modeling methods lack depth. Implementation details and hyperparameter fairness are not fully discussed.
4. Some theoretical results rely on restrictive assumptions, and their applicability to real-world data-generating processes is not well justified.

**Questions:**

see the weaknesses

---

> ### Author Response · Authors · 2025-11-19
>
> We sincerely thank the reviewer for their time and constructive comments, which have helped us improve the quality of our manuscript.
>
> Below is our response:
> 1. Quantifying noise is inherently difficult, and our method does not work on irregular cycles. We have also added a section under Limitations regarding multi-periodicity, as our approach is less advantageous when multiple equally strong periodicities are present.
> 2. To effectively mitigate distribution drift, we incorporate a combination of enhanced L2 regularization and dropout techniques during the model training process.
> 3. A key advantage of our approach is that it does not affect model architecture, ensuring the overall model size remains unchanged, thus there is no fairness issue.
> 4. Deriving a general theoretical result for all time series is complicated by the fact that our framework does not assume perfect periodicity. Therefore, the primary contribution of our theoretical analysis is to demonstrate the high effectiveness of our method specifically within the context of reduced rank regression.

---

### Meta-Review · Area_Chair_fVq6 · 2026-01-03

**Summary:**

The primary concerns include the lack of rigorous comparative baselines and the restriction of using the proposed method. While the concept of periodicity alignment is intuitive, reviewers highlight that the paper fails to compare against standard techniques for handling periodicity. Also, the method relies on a single, stable period which significantly limits its real-world applicability, as it might struggle with multi-periodicity, noise, and distribution shifts. Concerns were also raised regarding the unconventional 95%-5% train-test split without a validation set, and the choice of SFNN as the primary backbone, which is not a widely recognized benchmark in the time series domain.

**Reviewer Concerns:**

The authors clarified the use of z-normalization to justify zero-padding and added a Limitations section regarding multi-periodicity and non-periodic data. They also proposed using increased $L2$ regularization to combat distribution drift.

Many core issues remain unaddressed. Most notably, the authors refused to provide a comparison against periodic positional features, arguing that their method "maintains original architecture," which fails to address the reviewer's point about whether this "simple" method is actually better. The reliance on a single backbone (SFNN) remains a significant weakness, as the "model-agnostic" claim is not sufficiently shown across diverse architectures like PatchTST or DLinear. The rebuttal regarding the unconventional data split was insufficient, and the inherent contradiction between Theorem 4.2 and the paper's motivation also remains a point of concern.

**Reviewer Scores:**

Reviewer riNS (4 to lower)

Reviewer jEFr (2 to 2)

Reviewer kbzg (6 to 4)

Reviewer KvBP (6 to 6 or lower)

---

### Decision · Program_Chairs · 2026-01-26

Reject